# Production of High Silicon-Doped Hydroxyapatite Thin Film Coatings via Magnetron Sputtering: Deposition, Characterisation, and In Vitro Biocompatibility

**Samuel C. Coe [1], Matthew D. Wadge [1,*]**, **Reda M. Felfel [1,2], Ifty Ahmed [1], Gavin S. Walker [1], Colin A. Scotchford [1] and David M. Grant [1,*]**

[1] Advanced Materials Research Group, Faculty of Engineering, University of Nottingham, Nottingham NG7 2RD, UK; samuelccoe@gmail.com (S.C.C.); Reda.Felfel@nottingham.ac.uk (R.M.F.); ifty.ahmed@nottingham.ac.uk (I.A.); Gavin.Walker@nottingham.ac.uk (G.S.W.); colin.scotchford@nottingham.ac.uk (C.A.S.)

[2] Physics Department, Faculty of Science, Mansoura University, Mansoura 35511, Egypt

\* Correspondence: matthew.wadge@nottingham.ac.uk (M.D.W.); david.grant@nottingham.ac.uk (D.M.G.)

**Abstract:** In recent years, it has been found that small weight percent additions of silicon to HA can be used to enhance the initial response between bone tissue and HA. A large amount of research has been concerned with bulk materials, however, only recently has the attention moved to the use of these doped materials as coatings. This paper focusses on the development of a co-RF and pulsed DC magnetron sputtering methodology to produce a high percentage Si containing HA (SiHA) thin films (from 1.8 to 13.4 wt.%; one of the highest recorded in the literature to date). As deposited thin films were found to be amorphous, but crystallised at different annealing temperatures employed, dependent on silicon content, which also lowered surface energy profiles destabilising the films. X-ray photoelectron spectroscopy (XPS) was used to explore the structure of silicon within the films which were found to be in a polymeric ($SiO_2$; $Q^4$) state. However, after annealing, the films transformed to a $SiO_4^{4-}$, $Q^0$, state, indicating that silicon had substituted into the HA lattice at higher concentrations than previously reported. A loss of hydroxyl groups and the maintenance of a single-phase HA crystal structure further provided evidence for silicon substitution. Furthermore, a human osteoblast cell (HOB) model was used to explore the in vitro cellular response. The cells appeared to prefer the HA surfaces compared to SiHA surfaces, which was thought to be due to the higher solubility of SiHA surfaces inhibiting protein mediated cell attachment. The extent of this effect was found to be dependent on film crystallinity and silicon content.

**Keywords:** hydroxyapatite; thin film; RF magnetron sputtering; pulsed DC; Silicon

## 1. Introduction

One of the many issues faced with regards to foreign objects being inserted into the body is the reduction or elimination of adverse effects caused by a stimulated immune response. Many such effects can be reduced with a careful material selection; however the current trend is to use the response of the body to improve the initial success of implant materials which could lead to a longer device lifetime, reducing the need for revision surgery. In the case of load bearing implants, such as in total hip replacement (THR) or total knee replacement (TKR), further complications such as the mechanical requirements of the material to bear load can reduce the number of materials available. It is often the case that materials that possess the mechanical properties are not always the most compatible in

these sites. A possible solution is to coat mechanically sound devices with materials which will be more biologically favourable [1]. A considerable amount of work has already gone into this area [2,3] but many problems such as selection of the most appropriate correct coating technique is required to control properties such as coating adhesion [4], composition [5] and topography [6], which may ultimately influence the biological response of the surface.

While many material systems offer attractive properties for biomedical applications, HA has attracted much attention due to its excellent bioactive ability for the application of bone repair and bone replacement [7]. Unfortunately, the nature of ceramic materials due to their reduced mechanical performance in comparison to metals, means that they, including HA, are restricted to the application of non-load bearing sites [8]. This problem can be overcome by applying an HA coating onto metallic materials with superior mechanical strength and stiffness [9]. Currently, plasma spraying is the only suitable commercial and FDA approved coating process employed [10,11], but coatings are compromised by the inclusion of undesired calcium phosphate phases leading to bioresorption and variable tissue response [12]. In addition, these coatings are often poorly bonded to the metallic substrates, leading to failure in the form of cracking at the coating/substrate interface [13]. Physical vapour deposition (PVD), as an attractive alternative, offers dense, defect free, well adhered single phase coatings with controllable deposition parameters [14,15].

Numerous authors have showed RF magnetron sputtering could be a potentially beneficial technique for applying coatings to future implantable devices [16–20]. Current work has shown that HA coatings are bioactive and exhibit similar properties to bulk materials. Many different compositions have been achieved for HA thin films due to differing sputtering parameters [16–18,21]. However, it is still unclear what suitable deposition parameters are and how factors such as power density and sputtering environment affect resultant films. One important compositional parameter is the Ca/P ratio, which can give rise to changes in the thermal stability, biological response, mechanical properties and dissolution potential [17].

SiHA thin films have been shown to elicit an enhanced cellular response compared to HA, which may provide a way to increase bone response in vitro and in vivo, thus improving the longevity of implants for future generations [22,23]. Work concerning RF magnetron sputtered SiHA thin films have only recently been investigated in the last decade [19,23–26]. Thin film systems often differ from bulk material systems due to the manufacturing process. Therefore, it is assumed that silicate groups substitute for phosphate groups and it is not fully understood what may happen at higher silicon concentrations above 5 wt.% [23]. Furthermore, optimal silicon values (ca. 2.2 wt.%) have been suggested but these are not in agreement [27,28]. Therefore, higher silicon additions [29] up to 13.4 wt.% and altered crystallinities have been investigated in this study.

## 2. Materials and Methods

### 2.1. Substrate Preparation

A commercially pure grade 1 titanium sheet (Timet, Swansea, UK) (CPTi) was wire eroded into 10 mm diameter, 1 mm thick discs, which were then ground to a mirror finish ($R_a \sim 16 \pm 4$ nm) using a series of silicon carbide paper (P240-P1200; Struers, Rotherham, UK), polished with a mixture of colloidal silica (Buehler, Germany) and 10% hydrogen peroxide (Fisher Scientific, Loughborough, UK) and finally water. The discs were cleaned for 30 min in acetone, IMS and distilled water and dried under flowing nitrogen (BOC, London, UK).

Silicon single crystals (100) orientated Czochralski wafer (Compart Technology Ltd., Peterborough, UK) were cut into squares approximately 10 mm × 10 mm in size with a diamond tipped scribe, and were used as substrates for XRD to reduce peak contributions from the substrate.

## 2.2. Target Preparation

The HA targets were manufactured by plasma spraying HA powder (Plasma Biotal, Buxton, UK) onto a circular copper backing plate 75 mm in diameter and 5 mm in thickness, with a target material thickness of ca. 500 μm. For the Si doping, a single 99.999% pure silicon target (Kurt J. Lesker, Hastings, UK) was used, with dimensions 300 mm × 100 mm × 0.635 mm. Figure A2 demonstrates the target composition via XRD analysis to be mostly HA.

## 2.3. RF/Pulsed DC Magnetron Sputtering

The HA and SiHA thin film coatings were produced using a TEER UDP-650 type 2 unbalanced magnetron sputtering rig (Teer Coatings Ltd., Droitwich, UK). Before deposition, the chamber was pumped using successive rotary (Edwards E2M40 rotary pump; $10^{-2}$ Torr) and diffusion (Edwards 750 diffusion pump) pumping steps to a minimum of $2 \times 10^{-5}$ Torr. The chamber was then backfilled with argon (36 sccm; $2 \times 10^{-3}$ Torr; 99.999% purity; Pureshield BOC$^{©}$, London, UK).

The HA target was powered by an Advanced Energies RF power supply unit whilst the silicon target was powered by an Advanced Energy Inc. (Bicester, UK) DC Pinnacle pulsed DC power supply. The frequency and pulse time were maintained constant at 150 kHz and 1500 ns, respectively. A −30 V bias was applied to the substrates. The samples were mounted on a stainless steel plate with double sided adhesive tape and rotated at 4 RPM. The samples were bias-cleaned for 2 min prior to sputtering. The total deposition time for each coating type was maintained at 2 h, with approximate sputtering rates of ca. 100 nm/h.

## 2.4. Post Deposition Heat Treatments

The obtained thin film coatings were recrystallised via heat treatment at a range of temperatures, with a set time of 2 h using a Lenton Thermal Design tube furnace (Hope Valley, UK) in an argon atmosphere. Argon was flowed at 100 sccm for 30 min before ramping the temperature up to 20 °C min$^{-1}$. Mass spectrometry indicated that all gases and water vapour were at untraceable levels before ramping. The samples were then left to cool under flowing argon.

## 2.5. Materials Characterisation

### 2.5.1. Scanning Electron Microscopy (SEM) and Energy Dispersive X-Ray Spectroscopy (EDX)

A Phillips XL-30 scanning electron microscope (LaB$_6$, Surrey, UK) was used to obtain micrographs at accelerating voltages between 10–20 keV and a working distance of 10 mm. An Oxford Instruments energy dispersive X-ray microanalysis (EDX) system (High Wycombe, UK) was utilised with a collecting time of 50 s. Ten randomly selected spots were analysed. Approximately 90,000 total spectrum counts were taken per sample area.

### 2.5.2. Focused Ion Beam SEM (FIB-SEM) and Transmission Electron Microscopy (TEM)

The HA and SiHA thin films were initially sectioned using a FEI Quanta 200 3D FIB-SEM (Cambridge, UK) fitted with a Quorum cryo-transfer unit, an Omniprobe micromanipulator and an INCA Oxford Instruments EDX analysis system. The FIB-SEM was operated at an ion beam accelerating voltage of 30 kV and electron beam accelerating voltages of 5–20 kV. A coating of tungsten was deposited in situ by chemical vapour deposition (CVD) to protect the HA coating. FEI-Runscript software was employed to mill inspection trenches, using reducing milling currents of 7 to 1 nA for rough sectioning, followed by milling currents of 0.5 nA to 30 pA in order to polish the lamella surfaces and to fashion U-shaped cuts into the lamella to facilitate lift-out. A cross sectional microstructural and chemical analysis was performed using a JEOL 2000-FX-II TEM (Welwyn Garden City, UK) operating at 200 keV.

### 2.5.3. X-Ray Diffraction (XRD)

A Bruker AXS D8 Advance X-ray diffractometer (Coventry, UK) was used in glancing angle X-ray diffraction mode. Cu kα X-rays (λ = 1.5406 Å) incidented the samples at a 2θ range of 20°–55°, a step size of 0.02° and a dwell time of 11 s. The samples were mounted and rotated at 30 RPM. A 2θ range and dwell time of 25°–35° and 2 s, respectively, was used for recrystallisation measurements. Temperature scans were performed from room temperature to a maximum of 800 °C to investigate the recrystallisation of thin films.

PC-APD version 3.5B DOS-based software was used to determine crystallite size of thin films using the Scherrer equation:

$$B = \frac{0.9\lambda}{t\cos\theta} \tag{1}$$

where $B$ is the peak broadening (full width half maximum (FWHM) in radians), $\lambda$ is the wavelength of the XRD source material and $t$ is the crystallite diameter. B factors in instrumental broadening ($B = B_{0bs} - B_{inst}$) where $B_{obs}$ is the observed line broadening, which includes instrumental factors such as detector slit width, area of the specimen irradiated and possible $K_{\alpha 2}$ X-rays.

### 2.5.4. Reflective High-Energy Electron Diffraction

A RHEED unit coupled with a JEOL 2000fx TEM operated at 200 keV was used to assess the surface crystallinity of films. The samples were held perpendicular to normal on a stage positioned immediately below the objective lens. The samples were tilted so that the shadow edge was positioned close to the primary beam to access near surface diffraction information. A GaN/GaAs single crystal standard reference sample with known *d*-spacings and a camera constant of 33.8 ± 0.5 cm was used to calculate thin films' *d*-spacings.

### 2.5.5. X-Ray Photoelectron Spectroscopy (XPS)

A Kratos Instruments Axis Ultra (Manchester, UK) with a monochromated Al Kα X-ray source was run at 10 keV and 15 mA. The instrument was operated in constant analysis energy (CAE) mode with a pass energy of 20 eV for high-resolution scans. Chemical and compositional information was obtained between 0–1400 eV. All samples were charge corrected to the C 1s adventitious carbon peak, which was set to a value of 284.8 eV. The region areas were selected manually and peak deconvolution was carried out using Gaussian-Lorentzian (GL30) line shapes. The data analysis and compositional quantification were carried out using CasaXPS software (version 2.3.22).

### 2.5.6. Fourier Transform Infrared Spectroscopy (FTIR)

Fourier transform infrared spectroscopy (FTIR) was performed on the thin films using a Bruker Optics Tensor 27 spectrometer (Coventry, UK) in glancing angle mode at 80° with a liquid nitrogen cooled MCT detector. The chamber was flushed with compressed air continually at 500 sccm to reduce background signal from water vapour. Samples scans were taken in triplicate for each coating with a total of 90 scans per specimen. Resultant spectra were analysed using an OPUS software (version 7.0).

### 2.5.7. Surface Roughness – Optical Profilometry

A Mitutoyo Surftest SV-600 profilometer (Coventry, UK) equipped with Surfpak SV software was used to measure surface roughness of the coatings. The stylus had a 5 μm radius tip. A scan length of 2 mm with a scan speed of 0.2 mm s$^{-1}$ and a range of 80 μm were used for all samples. Calibration was carried out prior to every session using a Mitotoyo Precision reference specimen with an R$_a$ value of 2.95 μm.

### 2.5.8. Surface Wettability—Sessile Drop/Contact Angle

Wetting angles of water droplets on the sample surfaces were tested using a sessile drop experiment. $H_2O$ was pumped from a syringe at a rate of 1.0 $\mu L \cdot s^{-1}$, from a height 4.0 ± 0.5 mm above the sample's surface. A drop settling time of 10 s was implemented prior to data collection.

### 2.6. In Vitro Biocompatibility

#### 2.6.1. Alamarblue™ Assay

Duplicate samples were placed in 24 well plates (Nunc, Warrington, UK) and UV sterilised. Tissue Culture Plastic (TCP) was used as a control surface. Human Osteoblasts (HOBs) from bone chips of femoral heads of patients undergoing total hip arthroplasty were seeded into wells at a density of 40,000 cells $cm^{-2}$ and incubated at 37 °C and 5% $CO_2$. Media (Dulbecco/Vogt Modified Eagle's Minimal Essential Medium (DMEM) supplemented with 10% fetal bovine serum (Gibco Life Technologies, Inchinnan, Scotland), 1% L-Glutamine, 2% HEPES Buffer, 1% non-essential amino acids, 2% penicillin and streptomycin (Invitrogen, Rugby, UK) and 75 mg ascorbic acid (Sigma, UK)) were replenished every two days. At each respective time point (1, 4, 7, 10 and 14 days) the media were removed and samples were washed three times in phosphate-buffered saline (PBS) solution. Then, a 1 mL dilution of AlamarBlue™ (Serotec, Kidlington, UK) and Hank's balanced salt solution (HBSS) (Gibco, Inchinnan, Scotland) in the ratio of 1:10 was added to each well, including unseeded TCP wells, and incubated for 80 min. The well plates were subsequently wrapped in foil and shaken at 300 rpm on a Heidolph Titramax 100 for 10 min in a dark environment. The AlamarBlue™ solution was then removed and 100 $\mu m$ aliquots were transferred into a 96 well plate (Nunc, Warrington, UK). Fluorescence was measured using a Bio Tek Instruments FLx800 fluorescence plate reader (Swindon, UK) at 560 nm excitation and 590 nm emission filters. Unreduced AlamarBlue™ was subtracted from recorded values to remove background signal. The experiments were repeated twice.

#### 2.6.2. Alkaline Phosphatase Assay

After each designated time point, media were removed from culture plates and washed three times in PBS solution. Aliquots of 1 mL of sterile distilled water were added to each well. A freeze/thaw method was employed to lyse the cells. The samples were frozen at −20 °C and then allowed to defrost at room temperature. This was repeated three times. 50 $\mu L$ of lysate solution was added to a 96 well plate per sample which was mixed with 50 $\mu L$ of 4−nitrophenylphosphate (Sigma, UK) mixed with an appropriate quantity of diethanolamine buffer solution. Plates were shaken at 300 RPM for 1 min in a dark environment. The luminescence was then measured using a Bio Tek ELx800 luminescence plate reader with a primary wavelength of 405 nm and a reference wavelength of 630 nm.

#### 2.6.3. DNA Hoechst Staining Assay

At each time point media were removed and HOBs were washed within PBS three times and submerged in 1 mL of sterile distilled water. A freeze/thaw cycle was carried out in triplicate to lyse HOB cell walls. Then, 100 $\mu L$ of lysate was mixed with 100 $\mu L$ of Hoechst 33,258 stain (Sigma, UK) and shaken at 300 RPM for 1 min in a dark environment. Fluorescence was then read on a Bio Tek Instruments FLx800 fluorescence plate reader with 360 nm excitation and 460 nm emission filters.

#### 2.6.4. SEM Sample Preparation

At selected time points, media were removed and samples were washed with PBS thrice and replaced with 3% Glutaldehyde in 0.1 M sodium cacodylate buffer. This was replaced after 30 min with 7% sucrose solution in 0.1 M sodium cacodylate buffer. Specimens were then washed three times for 5 min periods with 0.1 M cacodylate buffer solution and then immersed in osmium tetraoxide for 1 h. Post fixing, cells were dehydrated using an ethanol/distilled water gradient (20% × 2 min, 40% × 5 min,

60% × 5 min, 80% × 5 min 90% × 5 min and 100% × 5 min × 2). Specimens were then submerged in hexamethyldisilazane (HMDS; Sigma, UK) for 5 min. This was then replaced with fresh HMDS and left to dry overnight. The samples were mounted on aluminium stubs with carbon adhesive tabs and gold/palladium coated (ca. 5 nm).

## 3. Results

### 3.1. Chemical and Structural Characterisation

#### 3.1.1. Film Morphology and Thickness

Figure 1A shows a cross-section of a HA thin film on a CPTi disc annealed at 600 °C. Selected Area Electron Diffraction (SAED) confirmed that the substrate and the coating was indeed CPTi and polycrystalline HA. Moreover, the coating can be seen to be uniform in thickness, free of voids or defects and 185 ± 4 nm in thickness. Figure 1B shows that the SiHA3 thin film measured 216 ± 5 nm in thickness; films became thicker with increasing silicon content. Local recrystallisation of the HA films was noted, however, this may be a result of the e-beam interaction.

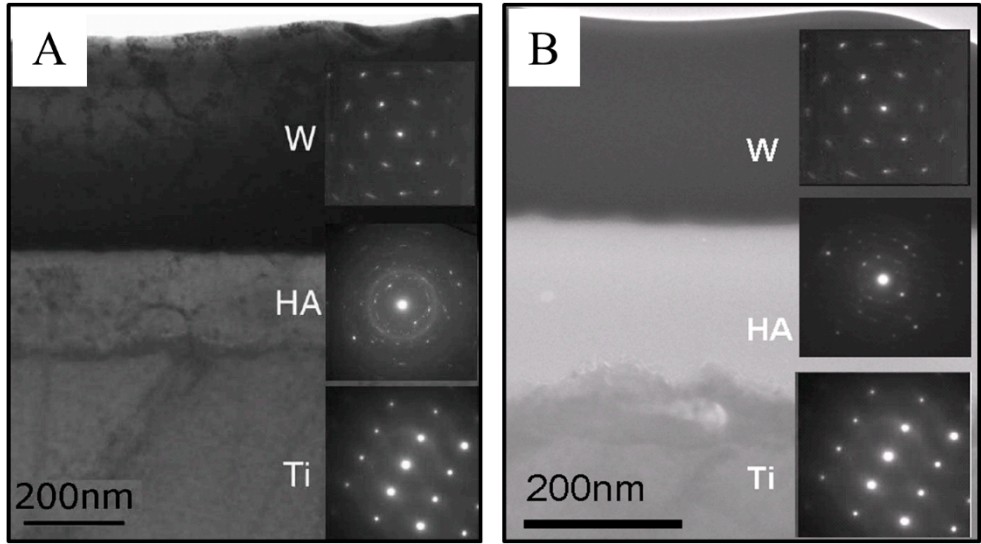

**Figure 1.** Bright field TEM image of (**A**) the tungsten/HA/Ti lamellar showing the HA thin film deposited on a CPTi substrate with associated selected area electron diffraction (SAED) patterns of the crystalline W protective coating, polycrystalline HA coating and crystalline Ti substrate and (**B**) the tungsten/SiHA3/Ti lamellar showing an amorphous SiHA3 coating on a CPTi substrate and associated SAED.

Surface micrographs of all films exhibited similar morphologies in as deposited and heat-treated states following the topography of the CPTi substrates. Figure 2A shows a representative as deposited HA film, showing a smooth dense coating without voids or defects. After a heat treatment at 600 °C (Figure 2B), all HA and silicon containing films looked similar in morphology with no notable differences with respect to silicon addition. At 700 °C, films became notably more textured with distinct features rising from the surfaces (Figure 2C).

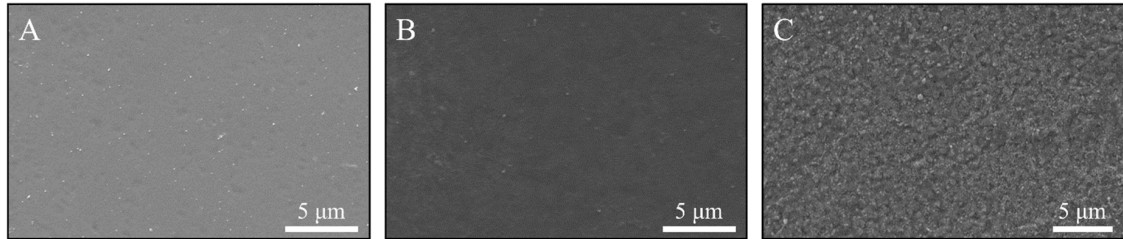

**Figure 2.** Representative SEM micrographs displaying the surface morphology of HA and SiHA thin films where (**A**) is an as deposited thin film on a CPTi substrate, (**B**) annealed at 600 °C in flowing argon for 2 h and (**C**) 700 °C in flowing argon for 2 h. Images taken from HA samples.

### 3.1.2. Energy Dispersive X-Ray Analysis (EDX)

Table 1 demonstrates averaged Ca/P ratios of all coatings, and plasma sprayed targets. All samples were significantly higher than the stoichiometric value of bulk HA (1.67), however, they were significantly lower than the Ca/P ratio of the target material. The standard error of the mean was found to increase with increasing silicon content. Silicon content was seen to increase from 1.8 to 13.4 wt.%, with increasing target power densities from $6.6 \times 10^{-4}$ to $3.3 \times 10^{-3}$, respectively.

**Table 1.** A summary of combined silicon content of HA and SiHA thin films batches as measured by EDX. Values displayed are the mean ± standard error of the mean, where $n = 6$. Ca/P content calculated from EDX and XPS measurements.

| Sample | Silicon Target Density/Wcm$^{-2}$ | Ca/P Ratio (EDX) | Ca/P Ratio (XPS) | Silicon Content/wt.% |
|---|---|---|---|---|
| **Stoichiometric HA** | N/A | 1.67 | 1.67 | N/A |
| **Plasma Sprayed HA Target** | N/A | 1.90 ± 0.02 | N/A | 0.0 ± 0.1 |
| **HA** | N/A | 1.76 ± 0.03 | 1.43 ± 0.03 | 0.0 ± 0.1 |
| **SiHA1** | $6.6 \times 10^{-4}$ | 1.74 ± 0.03 | 1.23 ± 0.05 | 1.8 ± 0.3 |
| **SiHA2** | $1.6 \times 10^{-3}$ | 1.79 ± 0.08 | 1.16 ± 0.06 | 4.2 ± 0.7 |
| **SiHA3** | $3.3 \times 10^{-3}$ | 1.68 ± 0.09 | 1.03 ± 0.13 | 13.4 ± 1.4 |

### 3.1.3. X-Ray Diffraction (XRD) Analysis

XRD spectra (Figure 3) show representative XRD plots of as deposited films onto silicon (100) wafers heat treated at both 600 (Figure 3A) and 700 °C (Figure 3B). All as deposited (unannealed) coatings revealed an amorphous nature with a distinct hump at 27.5°. Following heat treatment at 600 °C, HA, SiHA1 and SiHA2 films recrystallised forming a single-phase HA structure matching ICDD card 09-432. Preferential orientation was seen along the (002) reflection when compared to a randomly orientated sample. The peak intensity of films decreased after inclusion of silicon. This was seen by peak broadening along the (002), (211), (112) and the (300) planes. Surprisingly, the SiHA3 samples remained amorphous after heat treatment at both 600 and 700 °C. The silicon addition to SiHA3 clearly had an effect on the recrystallisation transitional temperature of the HA structure. Full recrystallisation was not achieved for the SiHA2 at 700 °C with clear broadening of the FWHM compared to the full crystalline HA.

Consequently, a sequential heat treatment investigation (Figure 3C) was conducted on the SiHA3 coating to determine the temperature of recrystallisation. Figure 3C shows that the structure of SiHA3 did not alter after heat treatment up to 700 °C. At 800 °C, a single phase HA structure matching ICDD card number 09-432 was observed. Preferential orientation along the (002) plane was no longer observed, with the (211) plane being the most intense. For further heat treatments up to 1000 °C, the intensity of the (002) increased, and peaks sharpened, indicating crystal growth.

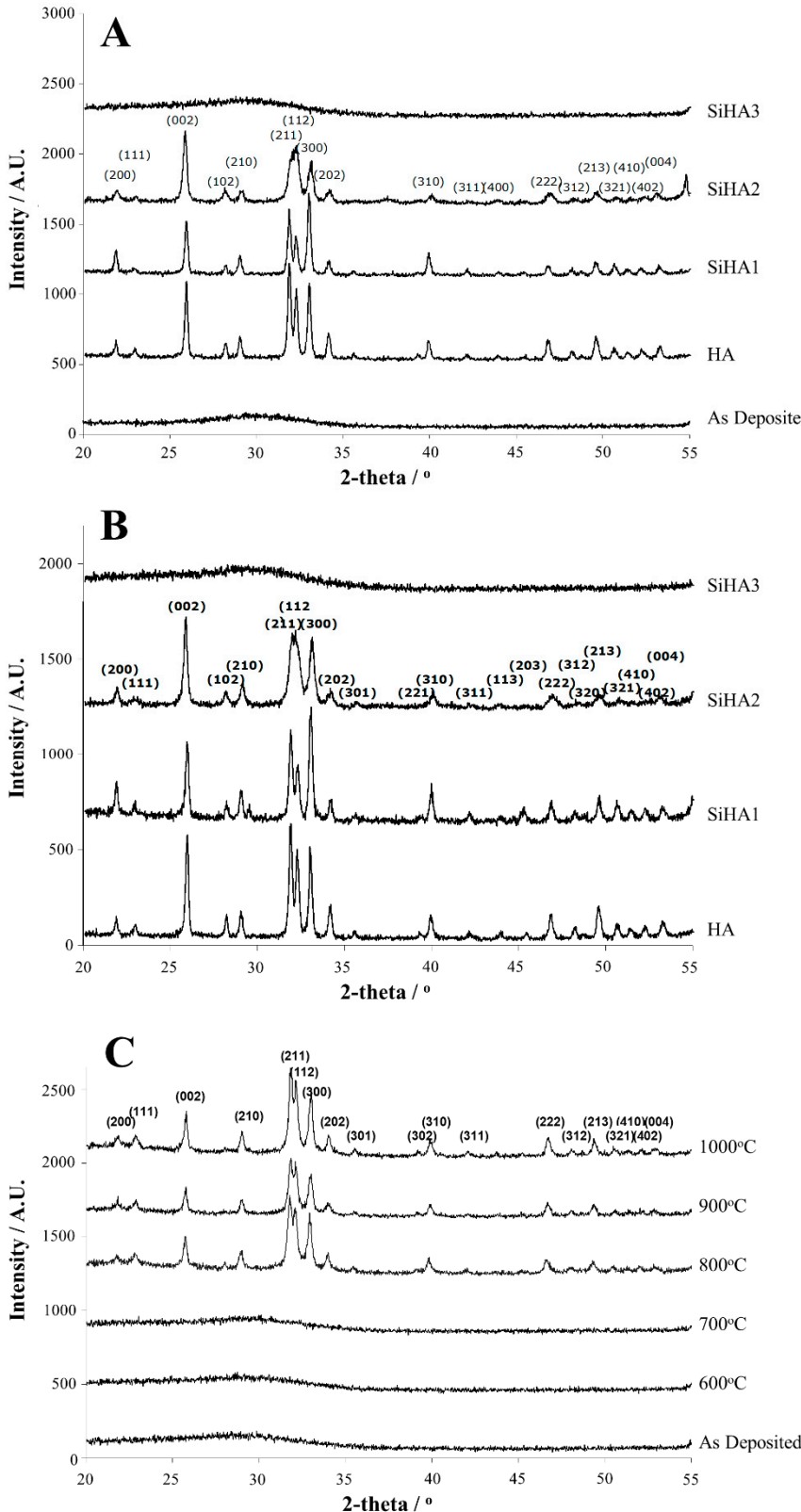

**Figure 3.** (**A**) and (**B**) Representative XRD plots of as deposited HA (unannealed), and HA and SiHA thin films on silicon (100) wafers heat treated at 600 °C and 700 °C in argon, respectively. (**C**) XRD plots of successive heat treatments on SiHA3 on silicon (100) wafer. Planes refer to ICDD 09-432 HA.

In addition, approximate crystallite size was calculated using the Scherrer equation. Table 2 lists calculated crystallite sizes of annealed HA and SiHA thin films from the XRD data presented. The crystallite size of all coatings increased with increasing annealing temperature. For example, the crystallite size of HA was ca. 78 and 89 nm following annealing at 600 and 700 °C, respectively. Respective values were seen to decrease with increasing silicon content. Crystallite size could not be calculated for SiHA3 at either annealing temperature, as both films were amorphous.

**Table 2.** A summary of HA crystallite size (nm) calculated by the Scherrer equation for HA and SiHA thin films sputtered onto silicon (100) single crystal wafer using the <001> planes.

| Sample | Heat Treatment Temperature/°C | |
| :---: | :---: | :---: |
| | 600 | 700 |
| HA | 78 ± 16 | 89 ± 16 |
| SiHA1 | 65 ± 14 | 71 ± 14 |
| SiHA2 | 71 ± 23 | 86 ± 31 |
| SiHA3 | N/A | N/A |

### 3.1.4. RHEED Analysis

All as deposited samples were amorphous (not shown), with RHEED being conducted mainly to ascertain crystallinity and phase purity. After annealing at 600 °C, diffraction rings could be observed (Figure 4A–D). Due to the low accuracy of the RHEED measurements and the large number of HA diffractions, it proved difficult to index the rings. Therefore, only the most intense rings have been indexed with confidence, with the first ring corresponding to the (002) plane and the second broader ring corresponding to the (211), (112) and (300) planes. With increasing silicon content, the number of rings present decreased. The SiHA1 sample, Figure 4B, exhibited only two hazy rings corresponding to the d-spacings at 2.9 and 2.7 Å. The first ring was assigned to the (002) plane and the second broader ring is a combination of the (211), (112) and the (300) planes. The SiHA2 samples, Figure 4C, displayed the same rings as above but with lower intensity. The SiHA3 samples, however, showed no rings, indicating that these samples were amorphous.

The samples annealed at 700 °C (Figure 4E–H) demonstrated sharper diffraction rings compared to 600 °C annealing. Ring intensity increased for the HA sample with no new observed peaks (Figure 4E). The SiHA1 sample (Figure 4F) detailed the presence of additional rings related to a combination of HA and rutile, which are shown in (Figure 5). For the higher silicon content coatings, (SiHA2 and SiHA3), the number of rings decreases, reverting back to a HA system, however, it may be seen that in SiHA2 (Figure 4G) some rings relating to rutile remain. This was also the case for the SiHA3 (Figure 4H) sample, however, the rings were more defuse.

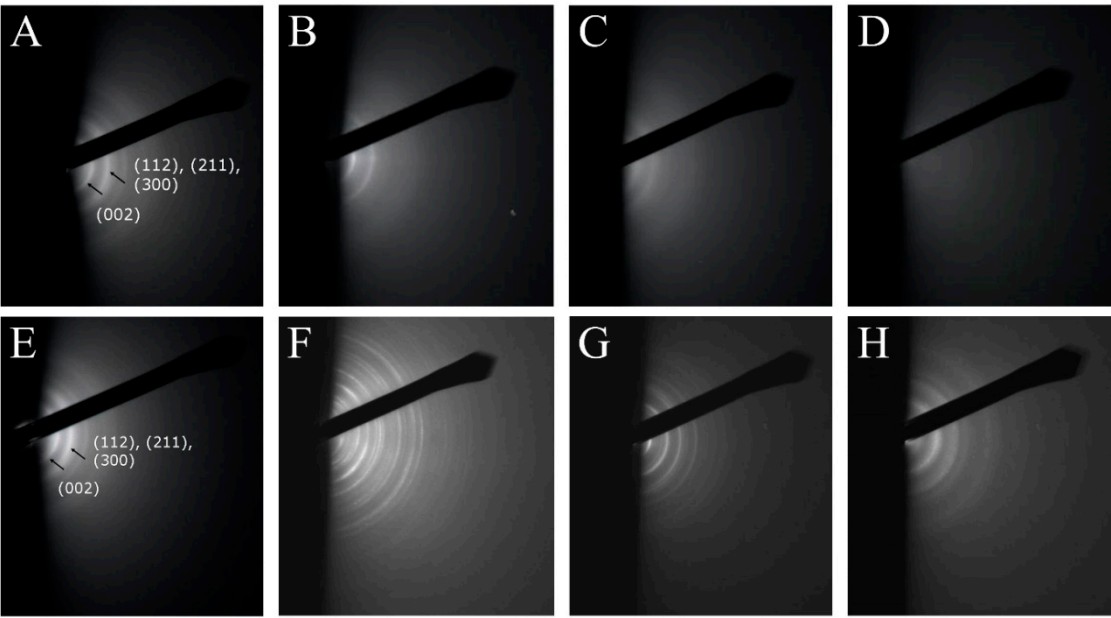

**Figure 4.** RHEED diffraction patterns of HA and SiHA thin films sputtered onto CPTi discs and annealed at 600 °C and 700 °C for A-D and E-H, respectively. Images were obtained at 200 keV. (**A**) and (**E**) HA, (**B**) and (**F**) SiHA1, (**C**) and (**G**) SiHA2 and (**D**) and (**H**) SiHA3.

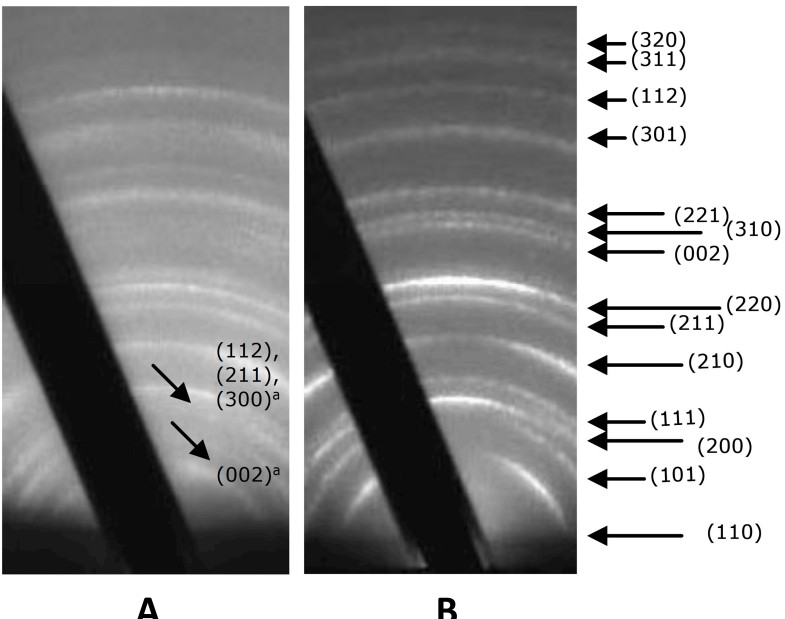

**Figure 5.** Comparisons of RHEED patterns for (**A**) SiHA1 film on a CPTi disc annealed at 700 °C in flowing argon for 2 h and (**B**) CPTi sample annealed in air at 750 °C for 1 h. All indexed planes match to ICDD card 76-1939 (Rutile) unless indexed with a superscript a plus diagonal arrows indicating possible HA reflections matching to ICDD card 09-432 (HA).

### 3.1.5. X-Ray Photoelectron Spectroscopy (XPS)

Ca 2p, P 2p and O 1s high resolution XPS spectra are shown in Figure 6. A calcium doublet was observed separated by 3.55 eV and fitted with two components at peak positions of 347.5 and 351.0 eV for Ca 2p1/2 and Ca 2p3/2 respectively [30]. Calcium was in low concentrations in the as deposited films (4.3–6.8 at.%), but increased after annealing at both temperatures (12.1–18.0 at.%). P 2p peaks were fitted with a doublet [31], with separation energy of 0.84 eV. Phosphorus content decreased with

both increasing annealing temperature and silicon content. Interestingly, no phosphorus was seen on any of the HA thin films annealed at 600 °C, but was seen on HA samples annealed at 700 °C.

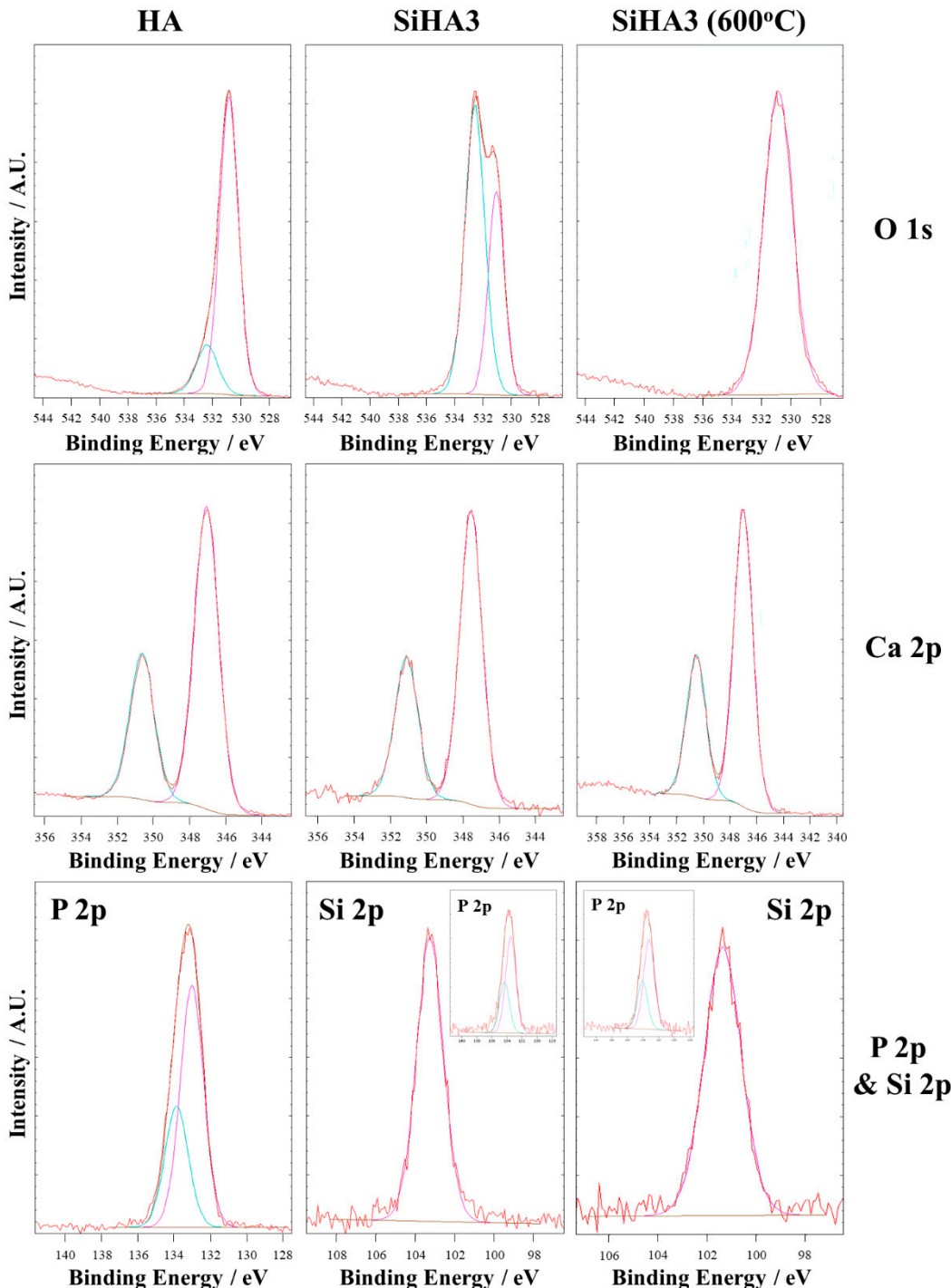

**Figure 6.** Representative high-resolution XPS spectra for HA, SiHA3 and 600 °C annealed SiHA3 samples, demonstrating the binding energy data for O 1s, Ca 2p, P 2p and Si 2p spectra, where appropriate.

The O 1s peak for the as deposited samples was fitted with two components at 531.0 eV and 532.5 eV, corresponding to $PO_4$ [32] groups and C–O [32] or $SiO_2$ groups [33], respectively. The second C–O component became larger with increasing silicon content. Therefore, it is thought that this is related to $SiO_2$ binding [33]. After annealing at 600 or 700 °C, the O 1s peak could only be fitted to a single component at 531.0 eV, which corresponded to $PO_4$ bonds. Furthermore, a reduction in oxygen content was seen in all films after annealing, however, no differences were observed between the oxygen content of films annealed at 600 or 700 °C. In the HA samples annealed at 600 °C (which showed no phosphorus present on the surface), no shift in the binding energy of the component at 531 eV was seen.

The Si 2p silicon peak was fitted to a single component. As already shown by the EDX data, the silicon content of thin films increased with increasing power density applied to the silicon target. These values, however, were in poor agreement with the EDX being consistently lower. In the as deposited samples the chemical shift for the Si 2p were found to depend on the silicon content of the film, with lower binding energies measured for the samples with lower silicon concentrations. After annealing at both temperatures all Si 2p peak positions were in the region of 101.5. Silicon content did not vary after annealing at 600 °C, compared with the as deposited samples, however after a heat treatment of 700 °C, only very small quantities were seen on all of the SiHA samples.

The Ca/P ratio decreased with increasing silicon content from 1.43 in the HA samples to 1.03 in the SiHA3 samples for the as deposited samples. After heat treatments of 600 °C the Ca/P ratio increased. This increase was higher for higher concentration silicon containing HA films. Following annealing at 700 °C a further increase in Ca/P ratio was seen.

### 3.1.6. Fourier Transform Infrared Spectroscopy (FTIR)

FTIR was used to assess the chemical bonding in RF magnetron sputtered thin films. Figure 7A shows infrared spectra for as deposited HA and SiHA thin films sputtered onto CPTi. The HA, SiHA1 and SiHA2 films exhibited four distinct bands at wavenumbers 1147, 1028, 950 and 617 cm$^{-1}$, which are indicative of $\upsilon_3$ P–O stretching. SiHA3 samples showed a reduction in the number of phosphate bands, with only peaks at 1147, 950 and 617 cm$^{-1}$ present.

After heat treatments at 600 °C (Figure 7B), the HA films showed sharper peaks with an additional phosphate band at 1080 cm$^{-1}$ when compared to the as deposited HA film. Moreover, a small OH peak was seen at 3643 cm$^{-1}$. The SiHA1 sample showed all phosphate bonds exhibited by the recrystallised HA sample, however, bands were slightly broader, with a new peak at 820 cm$^{-1}$. The SiHA2 sample showed broader phosphate bands, and the intensity of the peak at 820 cm$^{-1}$ was reduced. The SiHA3 sample only showed three broad phosphate bands at 1147, 950 and 617 cm$^{-1}$. This spectrum was very similar to the spectrum of the as deposited coating. Due to similarity of the produced spectra, the 700 °C heat treatment is not shown.

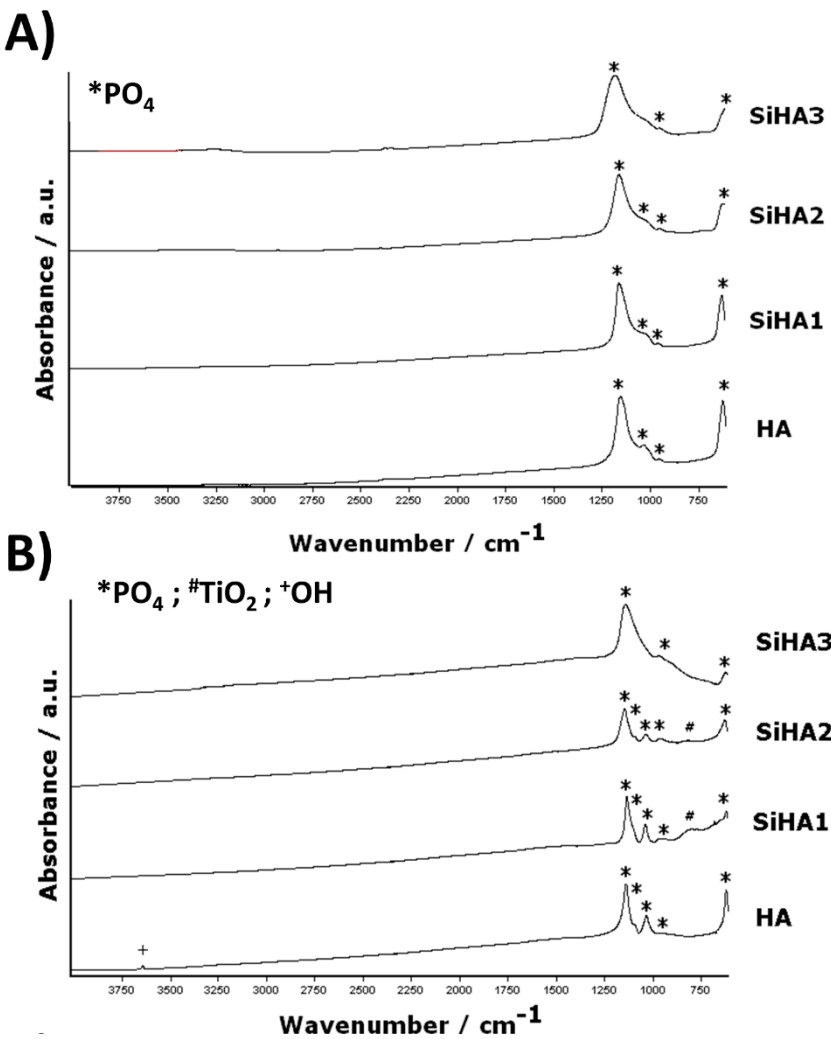

**Figure 7.** FTIR Spectra of (**A**) as deposited HA and SiHA thin films on CPTi substrates and (**B**) HA and SiHA thin films on CPTi substrates heat treated at 600 °C for 2 h in argon.

### 3.1.7. Surface Roughness

Stylus profilometry was performed on coatings to assess surface roughness and morphology. Figure 8 shows measured $R_a$ values for uncoated polished CPTi, HA and SiHA coated discs. Uncoated CPTi discs had an $R_a$ value of ca. 16 nm and their roughness increased to ca. 25 and 70 nm after heat treatment at 600 and 700 °C, respectively. As deposited HA and all SiHA coatings exhibited similar roughness, ca. 20 nm. After heat treatments at 600 °C, HA films had the highest roughness value at ca. 41 nm, which decreased gradually with increasing silicon content. This effect was also observed for all films heat treated at 700 °C, but at significantly higher roughness values than the samples treated at 600 °C, with HA films having a roughness of ca. 78 nm, and the SiHA3 samples being measured at ca. 32 nm.

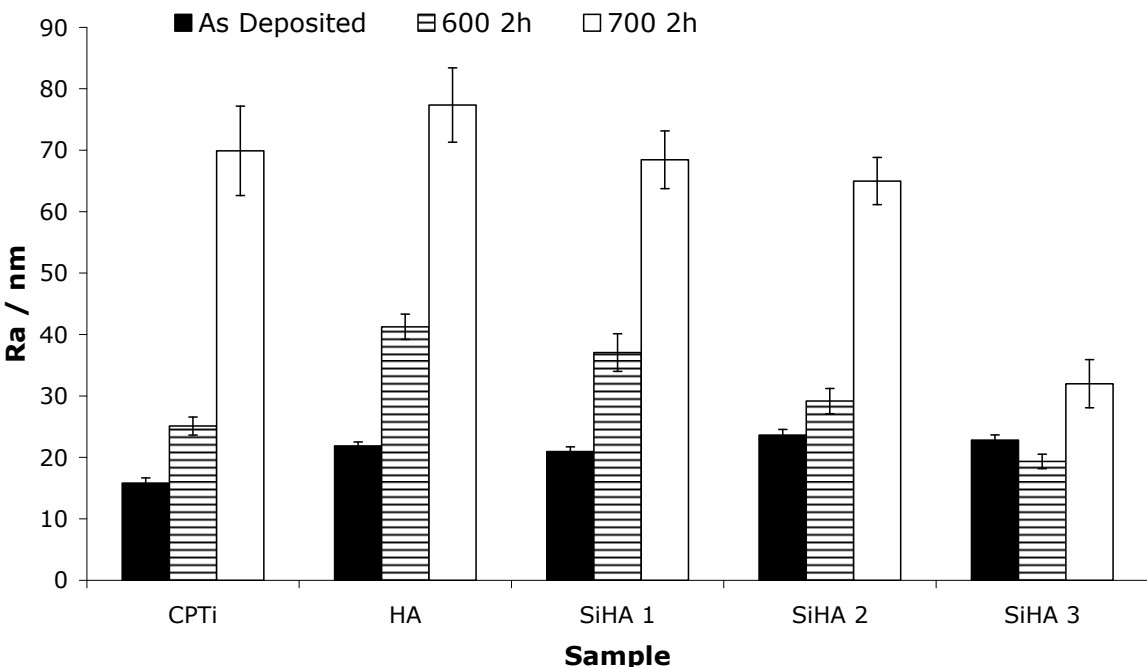

**Figure 8.** $R_a$ roughness values for CPTi, HA and SiHA thin films as received and heat treated at 600 and 700 °C in flowing argon for 2 h.

### 3.1.8. Film Wettability

Water contact angle testing was performed to establish the surface hydrophobicity or hydrophilicity of thin film samples. Figure 9 shows measured contact angles for HA and SiHA thin films. All silicon doped samples exhibited lower contact angles than the pure HA films whether in an as deposited or annealed state. The as deposited HA film had a water contact angle of 67°, which decreased with increasing silicon content down to an angle of 27° for the SiHA3 samples. Following heat treatments at 600 °C, the contact angle for all samples decreased when compared to the as deposited samples to values of 54, 41, 31 and 26° for the HA, SiHA1, SiHA2 and SiHA3, respectively. After heat treatments at 700 °C, an increase in contact angle was seen for all samples when compared to either the as deposited or samples heat treated at 600 °C, measuring 69, 56, 42 and 36° for the HA to the SiHA3 samples. Figure 9 also shows optical images of water droplets on as deposited HA and SiHA surfaces. It can therefore be seen that the hydrophilicity increases with increasing silicon content.

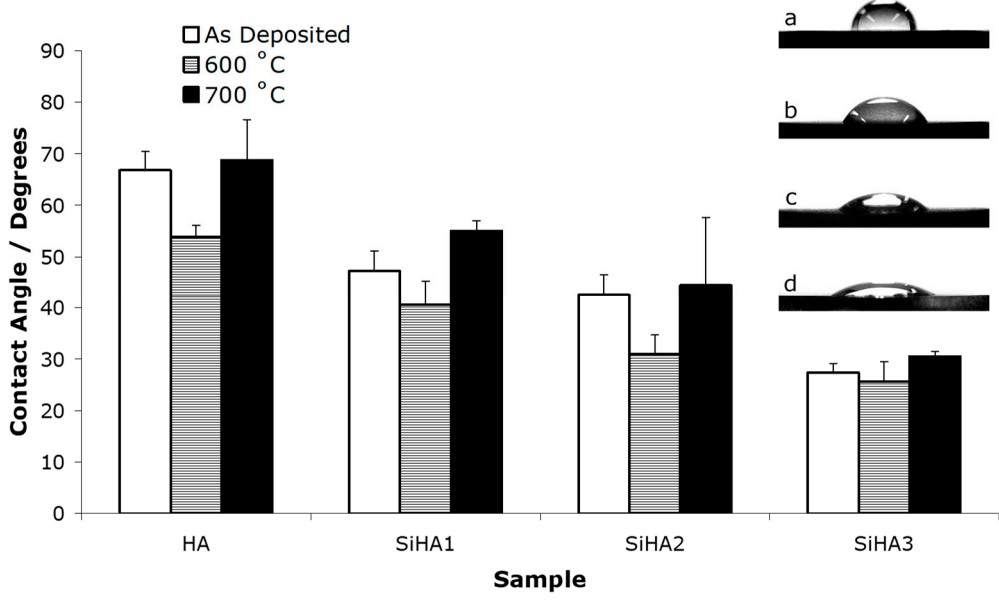

**Figure 9.** Measured contact angles of water droplets on HA and SiHA thin sputtered films. Mean ± standard error of the mean where n ≥ 6. Also shown are digitally enhanced representative grayscale photographs of water droplets on (**A**) HA (**B**) SiHA1 (**C**) SiHA2 and (**D**) SiHA3 thin sputtered films, showing the effect of silicon doping on the contact angle of water. Photographs taken from as deposited sample set.

## 3.2. In Vitro Cytocompatibility Testing

### 3.2.1. Elusion Testing – Metabolic Activity, DNA Content and Morphology

Figure 10A$_1$ shows no significant difference ($p > 0.05$) between the metabolic activity of either the control thermanox samples or the samples in HA and SiHA dissolution media. This was confirmed by the Hoest DNA staining assay (Figure 10A$_2$), in which there was no significant difference (p > 0.05) in DNA content of HOBs grown in the dissolution products of any of the coatings.

Cell morphologies for the control (Figure 10B insert) and all samples (Figure 10B–D) appeared to be similar, showing a monolayer over the thermanox surface. Cells were well spread, appearing to cover similar cell areas. Filopodia and lamellapodia were also observed, indicating cell signalling was occurring successfully and did not demonstrate significant cytotoxicity.

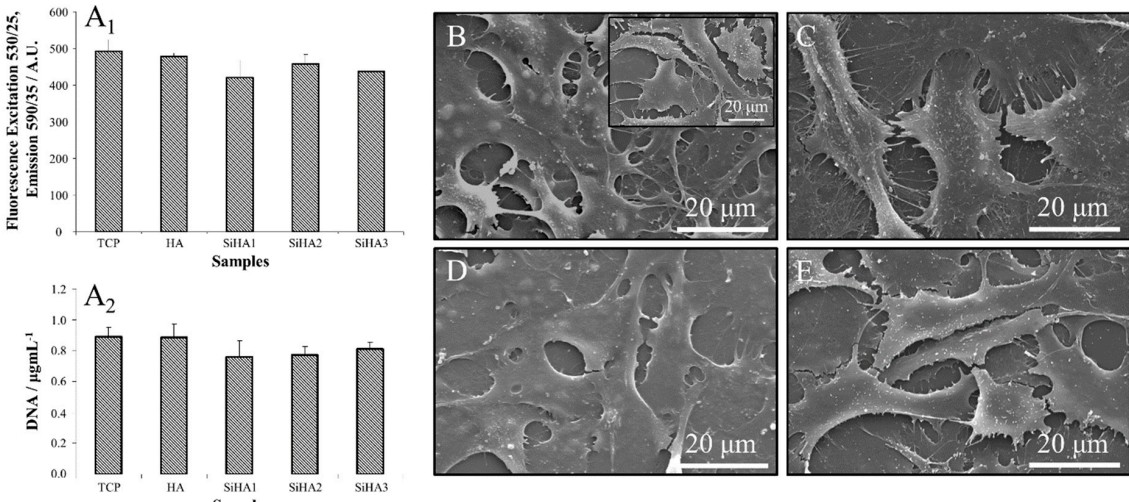

**Figure 10.** Combined elusion testing results showing: (**A₁**) Metabolic activity and (**A₂**) DNA content of pre-seeded HOB cells exposed to dissolution media of HA and SiHA samples for 24 h (Both expressed as mean ± standard error; n = 6); (**B**–**D**) SEM micrographs of HOB cell morphology on thermanox slides after 24 h of culture in media containing the dissolution products of (**B**) HA, (**C**) SiHA1, (**D**) SiHA2 and (**E**) SiHA3 thin films (insert image in B shows cells culture in fresh media for reference).

### 3.2.2. Initial Attachment

Figure 11 shows the adhesion of HOB cells as a percentage of a TCP control. As deposited and heat treated HA thin films showed good attachment after 90 min with values of 51 and 91% of the control, and were found to be significantly different ($p < 0.05$). As deposited and heat treated SiHA1 and SiHA2 samples showed poor cell attachment which was less than 20% of the control. All values for SiHA1 and SiHA2 samples were significantly lower than the heat treated HA thin film ($p < 0.05$). The as deposited SiHA3 samples showed good cell attachment compared to the as deposited and annealed SiHA1 and SiHA2 samples, and were not significantly different from the heat treated HA samples ($p < 0.05$). Following heat treatments at 600 °C for 2 h, the SiHA3 sample showed poor cell adhesion; 7% of the attachment seen on the control surface.

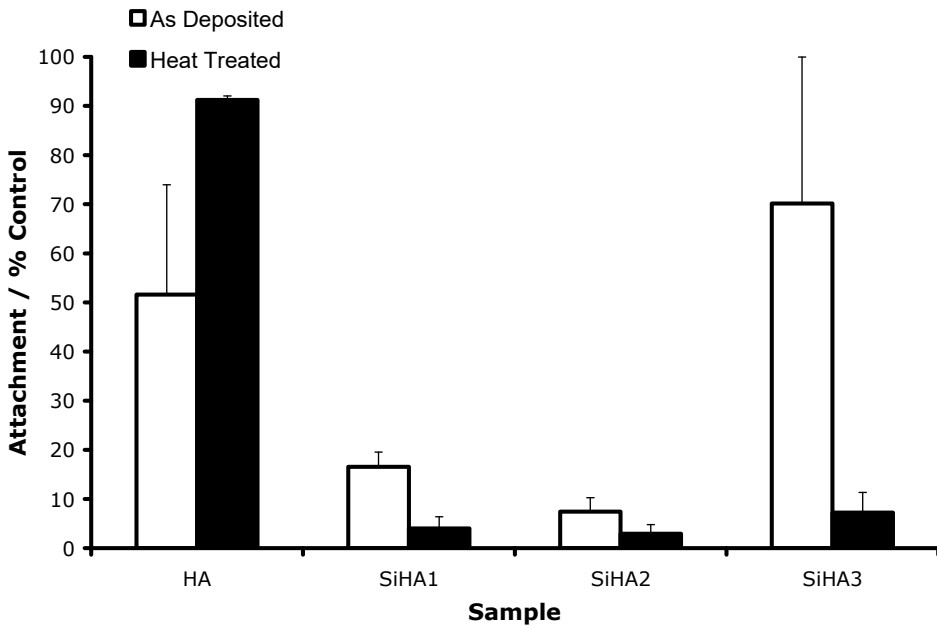

**Figure 11.** The 90 min attachment of HOB cells to as deposited and heat treated HA and SiHA films at 600 °C. Values are mean ± standard error where n = 6.

### 3.2.3. Proliferation and Differentiation—Cellular Activity, ALP, and Morphology

The AlamarBlue™ assay results of all sample types annealed at 600 °C (Figure 12A) demonstrate that the HA surface exhibited the highest cellular metabolic activity at every time point. The lowest activity was recorded on the SiHA1 samples. There appeared to be no increase in metabolic activity until day 14, where cell activity significantly increased. Conversely, metabolic activity increased with increasing silicon content of SiHA samples. The SiHA2 and SiHA3 samples showed higher cellular activity compared to the SiHA1 samples, and their metabolic activity increased gradually over time. However, HA and SiHA1 surfaces annealed at 700 °C (Figure 12B) demonstrated similar metabolic activity at all time points, but the differences were not statistically significant ($p > 0.05$). Overall proliferation of both samples was seen to increase gradually up to the 10 d time point, followed by a slight reduction at day 14.

DNA measurements (Figure 12C,D) confirmed the trends shown by the Alamar Blue™ assay, with HA annealed at 600 °C (Figure 12C) exhibiting the highest DNA content, followed by the SiHA3, SiHA2 and then the lowest, exhibited by the SiHA1 samples. HA consistently showed higher DNA values than all other samples. Annealing at 700 °C showed no significant difference between the HA and SiHA1 samples at any time point (Figure 12D).

ALP activity was negligible for all 600 °C annealed surfaces after seven days of culture (Figure 12E). After a time period of 10 days significant ALP activity was seen for the HA samples at approximately 35% of the TCP control. Negligible values were recorded for all silicon containing coatings. For the 700 °C annealed samples (Figure 12F), ALP was not expressed until day 10 for the HA sample, recording a value at 45% of the control with no significant difference at day 14. The SiHA1 samples arguably exhibited ALP production at day seven at approximately 10% of control. This was seen to significantly increase ($p < 0.05$) at day 10 to a value of 50%, which increased further to approximately 55% of the control at day 14.

The morphology of cells on thermanox slides (Figure 12G), HA (Figure 12H) and SiHA3 (Figure 12I) samples annealed at 600 °C at day 14 are shown in Figure 12G–I. SiHA3 was used as a representative for SiHA samples. HOBs on all surfaces appeared to be multi-layered, indicating desirable osteoblast cell growth. Both HA (Figure 12J) and SiHA1 (Figure 12M) surfaces (one day incubation) annealed at 700 °C appeared to be more textured than samples treated at 600 °C. Cells appeared to react to

this topography in the case of both samples by larger numbers of extending extra cellular processes compared to the 600 °C sample. After seven days of cell culture (Figure 12K,N), cells on both samples covered the sample surfaces and multilayering had occurred, and no differences were seen between the two samples. Cracks were also seen in both cell samples, which was due to the dehydration protocol adopted. Fourteen-day samples (Figure 12L,O) exhibited cracks again, but no difference in morphologies was seen. Some directional growth can be seen in both samples.

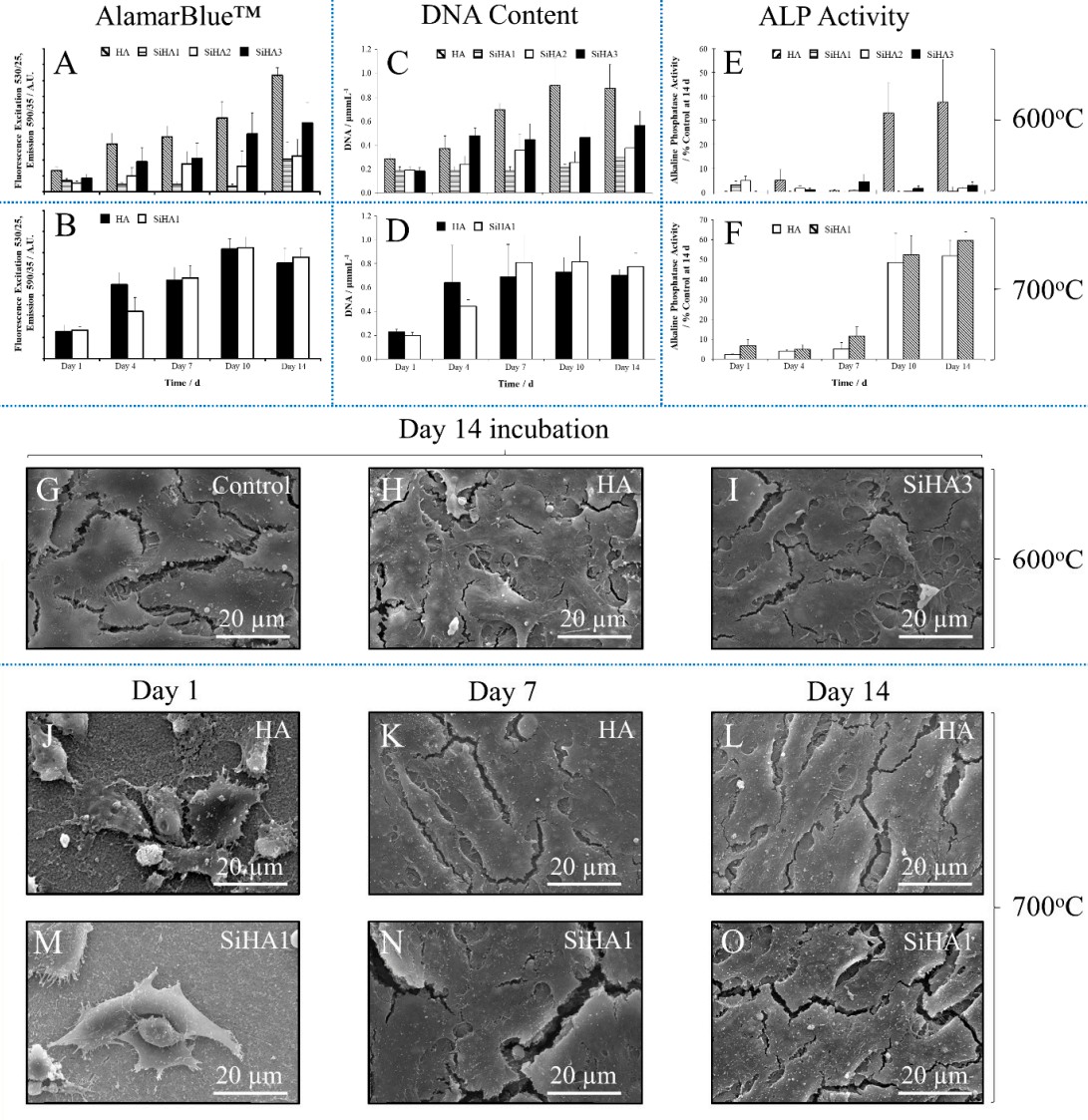

**Figure 12.** Combined cellular proliferation data showing: AlamarBlue™ assay of HOBs on HA and SiHA surfaces heat treated at (**A**) 600 °C and (**B**) 700 °C; DNA content of HOBs on HA and SiHA surfaces heat treated at (**C**) 600 °C and (**D**) 700 °C; ALP activity of HOBs on HA and SiHA surfaces heat treated at (**E**) 600 °C and (**F**) 700 °C (All graphs are plotted with mean ± standard error of the mean where n = 6); SEM micrographs of cellular morphology showing 14 day incubation on 600 °C annealed (**G**) Thermanox (Control), (**H**) HA, and (**I**) SiHA3; Further SEM micrographs of cell morphology on 700 °C annealed HA and SiHA1 samples incubated at day (**J**),(**M**) 1, (**K**),(**N**) 7 and (**L**),(**O**) 14, respectively.

## 4. Discussion

### 4.1. Composition and Topographical Analysis

Studies on bulk SiHA, as well as thick and thin film SiHA coatings, have identified them as eliciting an enhanced cellular and tissue response compared to HA [22,28,34–36]. The characterisation of thin film materials is still limited, despite detailed articles within the literature [25,37–45]. As a result, the role of silicon in the HA crystal structure is still not fully understood in the metastable system descried here and, moreover, how this interacts with cells; hence the importance and role of this study.

Morphologically, the surface (Figure 2) and cross-sectional (Figure 1) results, in conjunction with the XRD (Figure 3) and RHEED (Figures 4 and 5) analysis demonstrated clear trends regarding crystallisation of the films, in addition to increasing silicon content, which is in good agreement with Agyapong et al. [29] and Wang et al. [46] The films showed an increase in thickness with increasing power density applied to the Si target confirmed via TEM (Figure 1), which is to be expected since increasing the power density causes an increase in sputtering yield [47]. Furthermore, as is expected with magnetron sputtering deposition [30], the film exhibited good step coverage, as seen in the SEM (Figure 2) and roughness measurements (Figure 8). Silicon content had no effect on deposited film roughness but an increase in roughness was seen after annealing at 600 °C, except the SiHA3 sample, which remained constant. At higher annealing temperatures (700 °C), roughness typically doubled on all sample types; as crystals grow an increase in surface roughnesses will be observed. Furthermore, the higher the recrystallisation temperature the larger the crystallite size (Table 1) and in turn the more textured the surface (Figure 2). However, silicon inclusion has been shown by a number of authors to inhibit the crystallite growth of HA in both bulk and coatings [34], corroborating the current data, where roughness increases by a smaller amount for each increase in silicon addition to the point where the 13.4 wt.% SiHA shows no change in surface roughness (Figure 8).

For SiHA materials, it has been shown that structural configuration is important when trying to enhance osteoblast response [48]. Silicate ($SiO_4^{4-}$) is considered soluble, whereas silica ($SiO_2$) is insoluble in water, with Balas et al. [48] identifying that $SiO_4^{4-}$, due to enhanced solubility, generated a more favourable cellular response. XPS (Figure 6) found that binding energy shifts were consistent with increasing silicon content, with Okada et al. [49] and Stevenson et al. [50] confirming this trend. If fewer silicon atoms are present on the surface it is more probable that silicon will bond to oxygen, with higher silicon concentrations causing polymerisation; sharing electrons between silicon atoms. The increase in binding energy signifies a chemical change in Si–O bonding from polymeric $SiO_2$ ($Q^4$) to monomeric $SiO_4^{4-}$ ($Q^0$) structure; $Q^n$ where n represents the number of bridging oxygen atoms per $SiO_4$ tetrahedron. Post annealing (600/700 °C) exhibited Si 2p 3/2 binding energies of approximately 101.5 eV, suggesting a $Q^0$ structure, as detailed in two independent studies [48,51]; $SiO_4^{4-}$ had successfully substituted for $PO_4^{3-}$ groups. However, Balas et al. [48] demonstrated that this effect only occurred with up to 1.6 wt.% silicon addition in bulk materials, above which, it reverted back to a $Q^4$ configuration (103 eV). In the current study however, up to 13.4 wt.% (bulk)/6 at.% (surface) was successfully substituted into the HA films; a 'super saturated' state compared to theoretical values of 5 wt.% being substituted for $PO_4^{3-}$ tetrahedra in bulk SiHA [52]. The position of the silicate tetrahedra may occupy $PO_4^{3-}$ vacancies, however, further proof is required since OH site doping, leading to $Ca_{10}(PO_4)_4(SiO_4)_2$, could be possible, as no OH groups are seen in SiHA samples from FTIR analysis (Figure 7). However, the above phase would be detected in the XRD (Figure 3) and FTIR (Figure 7) data, but this was not the case, giving further evidence of a silicate substituted hydroxyapatite structure.

Ca/P ratios were significantly different from stoichiometric HA, with clear discrepancies between the EDX (bulk) and XPS (surface) data (Table 1), with EDX determining Ca/P ratios varied from 1.68 to 1.80, potentially from CaO formation [53]. Ratios obtained from XPS were all lower than EDX values (Table 1), likely due to preferential phosphate sputtering [54]. The Ca/P ratios of the as deposited samples decreased with increasing silicon from 1.43 to 1.03, potentially through the sputtering environment allowing the formation of P–Si–O bonds. This is a likely scenario as a large

number of Ca–P, Ca–O and P–O like species have been found in HA plasmas [53]. As more silicon is made available due to increasing bias on the silicon target, more P–Si–O groups may form, thus lowering the Ca/P ratio. Furthermore, it was seen that on annealing, the Ca/P ratio increased due to a loss of phosphorus [55,56].

From the FTIR data (Figure 7), increasing the silicon content caused widening of the $PO_4$ bands, suggesting bond formation is inhibited through silicon addition, even on annealed samples. The presence of OH bonding in the HA sample, which disappeared in the SiHA samples, is due to $SiO_4^{4-}$ species substituting for $PO_4^{3-}$ bonds; an imbalance of −1 is created. The most energetically favourable method of reducing this effect is to reduce the number of OH groups associated with the molecule [34,57]. The numbers of substitutions will be indirectly proportional to the number of OH groups. XPS (Figure 6) confirmed the super saturated state, which may explain why no OH was seen, opposed to just a reduction in OH peak intensity. This effect has also been shown in other apatite systems [58].

Wettability testing (Figure 9) showed that the incorporation of silicon into the HA lattice led to a more hydrophilic surface [51,59], demonstrating that SiHA has a more negative surface charge and increased surface adhesion than HA. Takeda and Fukawa [60] found that OH groups were a major factor governing surface chemical properties of oxide thin films. Higher contact angles were obtained for as deposited samples than samples annealed at 600 °C. When the samples were annealed, residual stresses could be corrected for, thus lowering the surface energy. This was confirmed by the small difference seen between the measured contact angles of as deposited and 600 °C SiHA3 samples, as this sample did not recrystallise at this temperature. After annealing at 700 °C, contact angles were higher than both the as deposited and 600 °C samples, likely due to the appearance of titanium and rutile phases at the sample's surface, as demonstrated from the RHEED analysis (Figures 4 and 5) [61].

As deposited HA and SiHA thin films were shown to be amorphous, with all samples except the SiHA3 sample showing a single phase HA structure post annealing (Figure 3); the SiHA3 samples required 800 °C annealing to recrystallise. Gibson et al. [34] and others have shown that introducing silicon into HA lowers the thermal stability. It has commonly been shown that silicon additions of 5 wt.% or more causes HA, on sintering, to decompose into undesirable phases like CaO and α- or β-TCP. However, no secondary phases were found in any of the films at any annealing temperature. This evidence, in conjunction with the XPS data (Figure 6), further suggests that higher amount of silicon may be substituted in the HA thin film structure than previously reported elsewhere. Crystallinity decreased with increasing silicon content, as confirmed by Zou et al. [43], for lower Si contents (0.8–2.0 wt.%). After annealing at 700 °C the SiHA1 samples showed rutile diffraction patterns, also confirmed via XPS (Figure 6) and RHEED (Figures 4 and 5).

### 4.2. In Vitro Cytocompatibility

Initial cell adhesion studies carried out on as deposited and HA thin films annealed at 600 °C demonstrated cells preferentially adhered to HA surfaces, with poor adhesion on all SiHA surfaces, with the exception of the SiHA3 as deposited samples, which showed good adhesion (Figure 11). Furthermore, the 14 day cell assays using HA and SiHA samples annealed at 600 °C provided further evidence that HOB cells preferred HA to the SiHA surfaces (Figure 12). Osteoblasts on HA surfaces showed increased proliferation but also were seen to be differentiating, indicated by the ALP activity. This affect was not seen on any of the SiHA surfaces after 14 days. This result was considered surprising, as a large amount of literature has been published demonstrating that SiHA ceramics lead to increased proliferation and differentiating activity of osteoblast cells with both bulk and coating materials in vitro and in vivo [62,63]. Assays ruled out that the samples had toxic effects on cells, moreover studies have shown that increased quantities of silicon in cell media can lead to the up-regulation of genes that aid cell proliferation and differentiation [64,65]. It is likely no enhancing effect was seen in our elution study because the test only exposed osteoblasts to media for one day (Figure 10). The stability of SiHA thin films in solution must be considered to be responsible for the poor adhesion and therefore the low proliferation compared with HA films. Qualitative EDX of samples used for the cytotoxicity study (see

Figure A1) confirmed that SiHA surfaces annealed at 600 °C are unstable; dissolving in cell culture media within hours for the highest doped sample. Cells have been shown to attach poorly to highly soluble (bioactive) surfaces, with the converse being noted for stable surfaces [66]. Despite cellular preference for HA surfaces, it was shown that cell number occurred in the order of SiHA3 > SiHA2 > SiHA1, from highest to lowest. When samples are immersed in cell culture media, serum proteins will attach to the surface allowing subsequent attachment of osteoblast cells. Over time, the film will dissolve, taking away with it attached adhesion proteins. Proteins will then change confirmation not allowing cells to attach. A new conditioning layer of protein will then redeposit but will again be removed by dissolution. If cells do manage to attach, they will subsequently be removed with the protein layer. When the protein changes confirmation, the cell will no longer be able to adhere and so will be released. This process will happen continually until cells can attach to a stable surface such as the underlying CPTi substrate. In the case of the SiHA1 compositions, this event does not occur until after day 10, but the dissolution rate was high enough to inhibit long term adhesion. SiHA2 however, showed increasing proliferation with time demonstrated by both the Alamarblue™ and the DNA assay (Figure 12). Again, from XPS data (not shown), only 1–2 at.% of the SiHA3 thin film remained after 2 days in cell culture media. Furthermore, it was observed in the contact angle testing experiments that some film dissolution would occur even when exposed to water for a few minutes. These observations and measurements may explain why initial adhesion of HOB cells is possible and sustainable on the as deposited and 600 °C SiHA3 films (Figure 11). Essentially, the CPTi substrates are revealed to cells which act as a stable protein mediated adhesion site. Initial adhesion studies comparing titanium and HA surfaces have demonstrated that titanium surfaces show a better response in a 90 min attachment period [67], but this was not seen in the case of the as deposited SiHA3 samples, which is thought to occur due to some cells undergoing apoptosis or programmed cell death during the prolonged attachment time. It is well known that cell adhesion via proteins allows signalling which can inhibit apoptosis [68]. Cell adhesion to the substrate via proteins is also necessary for a musculoskeletal cell's vitality, growth, migration, and differentiation [69,70].

It has been shown both in this study and other studies that SiHA bulk and thin film materials have a higher dissolution potential than HA [24,71]. Moreover, the staining protocol required for the initial adhesion may further affect the stability of the surface owing to numerous washing steps involved, accelerating film dissolution and removal of any adhered cells. In the current study, we investigated silicon contents higher than previously reported, ranging from 1.8–13.4 wt.%. Furthermore, coating thicknesses were higher, which tends to lead to higher residual stress in the films and on recrystallisation will give a higher crystallinity. In comparison to bulk materials, Arcos et al. [52] investigated the in vitro response of osteoblast cells to bulk high quantity silicon doped apatites. It was found that high silicon content apatite (low crystallinity) showed poor cell proliferation over a seven day period. This was explained by cells poorly adapting to their environments, however, it is more likely that this is due to surface dissolution inhibiting cell adhesion.

In order to overcome high dissolution rates of samples annealed at 600 °C, the cellular response of samples annealed at 700 °C were investigated, however, due to the reduced crystallinity and lower stability of the SiHA2 and SiHA3 samples, only the SiHA1 sample was investigated. Proliferation and differentiation on SiHA1 surfaces were slightly higher than on HA surfaces, however, this was not significantly different ($p > 0.05$). This conflicts with a large number of studies providing strong evidence that SiHA materials elicit an enhanced response when compared with HA materials [22–24,27,28,35,39,40,71–75]. This may be explained by the presence of a HA/rutile phases at the surface of the 700 °C samples. Moreover, a much lower silicon content was seen on the surface of the samples annealed at 700 °C compared to that of those annealed at 600 °C. Even so, it has been shown that even 0.4 wt.% silicon addition to HA can have a pronounced effect on adhesion and proliferation [27]. As almost no silicon content is present on the SiHA1 after annealing at 700 °C, it would be expected that this surface would have the lowest dissolution rate and be unlikely to cause problems for cell adhesion, but it may in fact have a beneficial effect leading to the slightly

increased cellular response, although this was not shown to be significantly different (p > 0.05) from the HA sample.

Commonly in the literature, it has been seen that increasing the post deposition temperature of HA ceramics increases the cell proliferation and differentiation in both bulk and thin film systems [67,76,77]. Roughness, topography chemistry and surface energy are all known to influence cell response to a given surface [78]. Data obtained would suggest that in the current study cells have reacted to the roughness and chemistry. The majority of studies concerned with topography have concentrated on the micron scale, with only a few authors concentrating on the nanometre scale. This is mainly due to a lack of knowledge of how to produce such surfaces, however Affrossman et al. [79] have used polymer demixing to achieve nanotopographies. It has been reported that cells can detect changes as small as 5 nm and in vivo cells commonly respond to 66 nm banding on collagen fibrils [80]. In this study there was a roughness difference of 35 nm and cells were shown to react to this will increasing numbers of lamellapodia and filopodia leading to distinct attachment sites. This led to no significant difference (p > 0.05) in cell number and metabolic activity, suggesting that roughness values on this scale have no major effects on cellular response. Recently, Kahn et al. [81] used neural cells to investigate several surface textures ranging from 10 to 250 nm in roughness. Values between 20–100 nm promoted cell adhesion and longevity, however, surfaces led to a decrease in attachment at values > 100 nm. Similar trends have been found in other studies using different cells [82], but it is often the case that differences as low as 30 nm did not yield any notable difference. Dalby et al. [83] studied the effect of nano-islands on polystyrene materials with fibroblasts. It was found that islands as low as 13 nm high led to increased adhesion, proliferation and cytoskeletal development when compared to flat controls. Conversely, nano-islands 95 nm in height lead to unusual, stellate morphologies with poorly formed cytoskeletons [84]. Intermediate islands (45 nm) showed no difference in cell area from the control, however the cytoskeleton was less well formed. Studies have shown that RF magnetron sputtered HA surfaces show no significant difference (p > 0.05), when compared with titanium substrates at initial time points [28,85] and the current study agrees with such work. It may, however, be that because phosphorus was not found at the top few atomic layers the cellular response was impaired. While not directly comparable, it has been shown that cells respond preferentially to surfaces with stoichiometric Ca/P values [86]. The literature confirms that surface texture and chemistry are important, but it is still under debate which has a more positive effect.

Overall, the combination of nanotopography and change in surface chemistry has led to small changes in cell morphology and proliferation over a 14 day time period, however such differences in the HA and SiHA1 surfaces annealed at 700 °C for 2 h were too subtle to be significantly different.

## 5. Conclusions

The work performed in this study investigated HA and SiHA RF/Pulsed DC magnetron sputtered thin films as coatings for orthopaedic applications. As deposited HA thin films were found to be amorphous or nanocrystalline, however, upon annealing (600 °C for all samples, except SiHA3, which required 800 °C) recrystallised. Furthermore, the addition of silicon to HA thin films inhibited HA crystallite growth as demonstrated by crystallite sizes calculated from XRD line broadening.

Both EDX and XPS showed a reduction in Ca/P ratio with increasing silicon content due to possible creation of a P–Si–O chemical species during deposition, which allows P to reach the substrate more readily. After annealing, however, the Ca/P ratio increased with increasing temperature, likely due to the evaporation of volatile phosphate species, facilitated by silicon inclusion destabilising the HA structure. XPS demonstrated that deposited SiHA thin films contain polymerised silicate networks, transforming to monomeric states after annealing, suggesting $SiO_4^{4-}$ substitution in the HA lattice for $PO_4^{3-}$ chemical species.

The roughness was shown to be similar for all as deposited films, which were measured to be approximately 20 nm (Ra), similar to the CP-Ti substrate, which increased following annealing; this was inversely proportional to the silicon content, due to silicon inhibition of the HA crystallite growth

and the rise of rutile grain growth. Silicon is also known to inhibit rutile growth, thus explaining the lowering in roughness with increasing silicon content. Water contact angle testing demonstrated silicon addition to the HA structure increased hydrophilicity with increasing silicon content.

Initial adhesion, proliferation and differentiation assays all suggested HOBs preferred HA to the SiHA surfaces, due to silicon doping destabilising the HA thin films, removing the protein conditioning layer essential for normal cell adhesion and growth. HOBs on the highest silicon doped HA thin films annealed at 600 °C showed some proliferation due to the stable CPTi substrate surface becoming available for protein mediated cell adhesion. After annealing at 700 °C, no significant difference (p > 0.05) was seen between the HA and the SiHA1 surfaces, suggesting enhanced cellular response due to crystallinity levels.

This study ultimately demonstrates that for higher (one of the highest tested in the literature), meta-stable doping levels of Si into the HA structure, cellular response is strongly linked to the crystallinity of the produced HA and SiHA films, the surface stability, as well as other properties, such as surface wettability, roughness, etc. Despite literature studies showing small doping levels of Si have a positive effect on cellular proliferation, this is not seen in higher-doped systems and, therefore, careful optimisation is required to glean appropriate properties.

**Author Contributions:** Conceptualization, S.C.C., G.S.W., C.A.S. and D.M.G.; methodology, S.C.C., G.S.W., C.A.S. and D.M.G.; validation, S.C.C., M.D.W., R.M.F., I.A., G.S.W., C.A.S. and D.M.G.; formal analysis, S.C.C., and M.D.W.; investigation, S.C.C.; resources, S.C.C., G.S.W., C.A.S. and D.M.G.; data curation, S.C.C. and M.D.W.; writing—original draft preparation, S.C.C., M.D.W., R.M.F., and D.M.G.; writing—review and editing, S.C.C., M.D.W., R.M.F., I.A., G.S.W., C.A.S. and D.M.G.; visualization, S.C.C., M.D.W., R.M.F., I.A., G.S.W., C.A.S. and D.M.G.; supervision, G.S.W., C.A.S. and D.M.G.; project administration, G.S.W., C.A.S. and D.M.G.; funding acquisition, G.S.W., C.A.S. and D.M.G. All authors have read and agreed to the published version of the manuscript.

**Funding:** This work was supported through funding from Teer Coatings Ltd.

**Acknowledgments:** The authors would like to thank all technical staff who aided in equipment operation, particularly Keith Dinsdale, George Anderson, Martin Roe, Nigel Neate, Julie Thornhill, Tom Buss, Rory Screaton, and Graham Malkinson.

**Conflicts of Interest:** The authors declare no conflict of interest.

## Appendix A

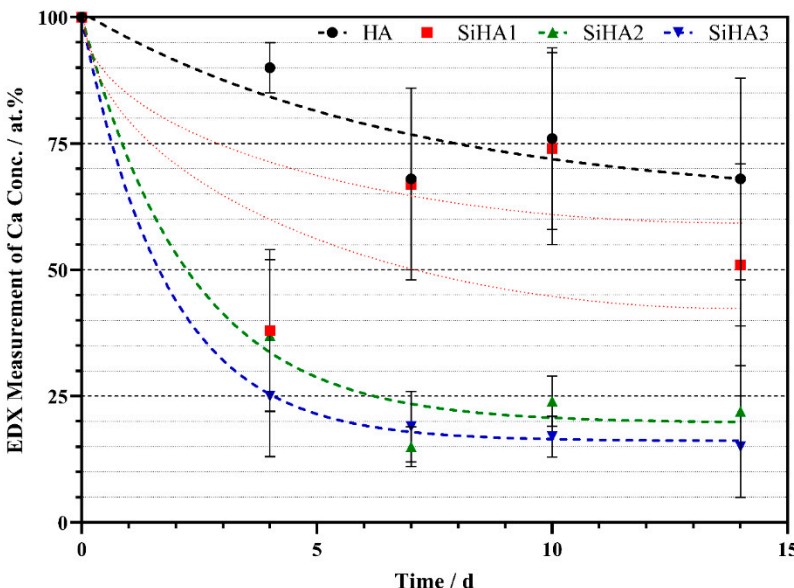

**Figure A1.** Percentage of HA and SiHA thin films annealed at 600 °C remaining as a percentage of original coating Ca content. Data plotted is mean ± standard error of the mean; n = 4. Non-linear one-phase decay regression plots were calculated using GraphPad Prism software based on the data shown. Due to the high variance of the SiHA1 sample, a suitable regression line was unable to be plotted, hence a probable area has manually been fitted for visual enhancement.

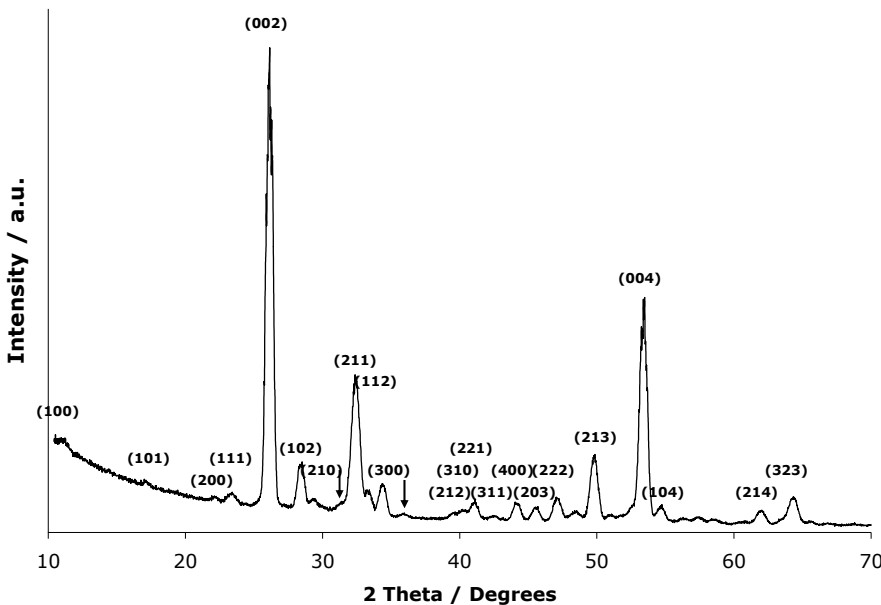

**Figure A2.** XRD plot for a Plasma Biotal plasma sprayed copper backed target. All major diffraction planes are indexed. Arrows indicate potential β-TCP secondary phase, however, with the low intensity accurate quantification is difficult.

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
