# Peer review of "Production of High Silicon-Doped Hydroxyapatite Thin Film Coatings via Magnetron Sputtering: Deposition, Characterisation, and In Vitro Biocompatibility"

_coatings, doi:10.3390/coatings10020190_

Round 1

Reviewer 1 Report

The referred article is interesting and high-level investigation representing results of comprehensive study for HA-based coatings obtained by PVD (magnetron sputtering). Special attention has been payed to investigation of silicon effect on the coating properties. Authors used a lot of experimental methods to obtain information about the coating physical properties and their effect on osteoblast cells that was studied in vitro. In my opinion, the article may be published in the “Coatings” journal, however, before publishing the article should be subjected to minor revision and the following questions should be answered:

Authors describing existing methods for calcium HA-based coating deposition indicated quite right in lines 49-52 that ”…plasma spraying is the only suitable commercial … coating process employed …, but coatings are … compromised by the inclusion of undesired calcium phosphate phases leading to bioresorption and … variable tissue response”. Besides, another very serious drawback for the plasma spraying (PS) method indicating in lines 52-53 is low adhesion of the coating to metal substrate. However, as information for this article authors I would like to note that method of gas detonation deposition (spraying) as an alternative to PS allows one to achieve extremely high adhesion, good crystallinity of the HA-based coatings and possesses other advantages described in literature. In section 2.2 “Target preparation” (lines 84-86) authors described HA target as thick HA layer obtained onto Cu substrate by PS method. But the layer may be strongly disordered and contain other phases like TCP, TTCP, CaO (see, for example [S.W.K.Kweh et.al. High temperature in-situ XRD plasma sprayed HA coatings. Biomaterials, 23(2002), 381-387]) those may influence properties of the films obtained using such target. Thus, it is not “HA target” but composite one. Besides, the only Ca/P ratio for the target is given in the article (page 7, Table 1) and XRD data for the target is absent. Purity of all gases used in technological processes must be indicated. It is especially important for PVD because of, for example, oxygen in balloon gases can influence film properties. What is deposition time (or deposition rate) for the films? The only time for sample treatment prior to deposition is given (line 98). What about substrate temperature during deposition, is a substrate cooled during the processes? In Table 1 (Page 7) the Ca/P ratios for stoichiometric HA measured by EDX and XPS methods coincide but for HA film even without silicon significant discrepancy in the values was observed. May it be connected with target properties (see question 2)? XRD results description (page 7, section 3.1.3) differs from caption to Fig. 3 (page 8). Captions for Fig. 3A and Fig. 3C is confused. What does it mean “As deposited” curve on Fig. 3A? Is it XRD spectrum of any unannealed film submitted on Fig. 3A? It seems that in accordance with balance requirement it should be Ca10(PO4)4(SiO4)2 instead of Ca10(PO3)4(SiO4)2 (line 497).

Reviewer 2 Report

The finding of this work was interesting and very useful in the field of biomaterials and biomedical. However, the following concerns should be addressed before accept.

Although a large number of references have been cited in this manuscript, many of them were rather outdated. Recent research reports should be cited, especially when introducing RF magnetron sputtering and SiHA thin films, the latest research progress should be mentioned. Page 2, line 71 “Therefore, higher silicon additions up to 13.8 wt %...”. It should be “13.4 wt %”. Page 14, line 375 “It can be seen that the hydrophobicity increases with increasing silicon content.” The conclusion was incorrect. As for the surface roughness and micromorphology analysis, why not use AFM characterization? The AFM images should be more vivid. Figure 10, for convenience comparison, it should better display the inserted image (reference) separately. Figure S1 was a key data to indicate the “the higher solubility of SiHA surfaces inhibiting protein mediated cell attachment.” Thus, I suggested placed in the text. Besides, if possible I wonder to know what’s the dissolved substance and their content? Cell attachment usually depended on the surface composition, morphology, and wettability. Have the authors thought about the influence of wettability (change in hydrophilicity) on cell attachment? The section of the conclusion needs to be more concise.

Reviewer 3 Report

The paper "Production of High Silicon-doped Hydroxyapatite Thin Film Coatings via Magnetron Sputtering: Deposition, Characterisation, and In Vitro Biocompatibility" by Coe et al. displays a thorough description of preparation and characterization of biomaterials for implantation.

The paper needs slight improvement, since some of the materials used in the study are  not defined clearly at their first ocurence in the text. For example the term HOBs (I suppose it means Human Osteoblasts) is not explained as such; the same for TCP. Moreover, it is recommended to indicate the source of cells used in the experiments, as well of other materials used.  With regard to the in vitro assays, I suggest the authors to insert information regarding materials used in the culture experiments (such as plates, since, for instance, using a different brand could affect cell behavior); the source of culture medium should be indicated, as well as for the eventual supplements used; I suppose they used a culture medium, eventual supplements and also some serum (like fetal bovine serum); these need to be clearly described (indicating the providers).
